# Cryogenic-Compatible Spherical Rotors and Stators for Magic Angle Spinning Dynamic Nuclear Polarization

Lauren E. Price[1], Nicholas Alaniva[1], Marthe Millen[1], Till Epprecht[1], Michael Urban[1], Alexander Däpp[1], Alexander B. Barnes[1]

[1]Department of Chemistry and Applied Biochemistry, ETH Zürich, Zürich, 8093, Switzerland

*Correspondence to*: Alexander B. Barnes (alexander.barnes@phys.chem.ethz.ch)

**Abstract.** Cryogenic magic-angle spinning (MAS) is a standard technique utilized for Dynamic Nuclear Polarization (DNP) in solid state nuclear magnetic resonance (NMR). Here we describe the optimization and implementation of a stator for cryogenic MAS with 9.5 mm diameter spherical rotors, allowing for DNP experiments on large sample volumes. Designs of
the stator and rotor for cryogenic MAS build on recent advancements of MAS spheres, and take a step further to incorporate sample-insert/eject and temperature-independent spinning stability of +/- 1 Hz. At a field of 7 T and spinning at 2.0 kHz with a sample temperature of 105-107 K, DNP enhancements of 256 and 200 were observed for 124 μL and 223 μL sample volumes, respectively, each consisting of 4 M $^{13}$C, $^{15}$N-labelled urea and 20 mM AMUPol in a glycerol-water glassy matrix.

## 1 Introduction

Dynamic Nuclear Polarization (DNP) is a method that increases sensitivity in nuclear magnetic resonance (NMR) through transfer of electron-spin polarization to coupled nuclear spins (Hu et al., 2004; Lilly Thankamony et al., 2017; Afeworki et al., 1993). This orders-of-magnitude improvement enables the investigation of otherwise unobservable systems in fields such as biology (Albert et al., 2018; Overall et al., 2020; Hirsh et al., 2016) and material science (Shinji Tanaka et al., 2022; Venkatesh
et al., 2020) and yields greater experimental throughput (Smith and Long, 2015). Pivotal to the performance of DNP in solid-state NMR is stable cryogenic magic-angle spinning (MAS). Recently, spherical rotors for MAS were introduced, providing novelty and flexibility in the MAS apparatus design while maintaining robust spinning performance (Chen et al., 2018). Here, we utilize these qualities of the spinning apparatus, or stator, to extend the applicability of MAS spheres to cryogenic MAS for DNP.

Commonly employed DNP mechanisms in solid-state NMR rely on the relatively long relaxation times of unpaired electron spins at "cryogenic temperatures" (typically below 120 K), in combination with applied microwaves (Scott et al., 2018b; Gao et al., 2019b; Nanni et al., 2013; Barnes et al., 2012), to facilitate the transfer of polarization. As electron spins are more highly-polarized than nuclear spins, this serves to improve the sensitivity of the observed nuclear spin signal. Improved resolution in solid-state NMR is made possible by MAS, which averages anisotropic nuclear spin interactions (Cohen et al., 1957; Andrew

et al., 1958; Andrew, 1981). The conventional technique for MAS utilizes a cylindrical sample chamber/rotor and two sets of gas to support and spin the sample rotor, which features a turbine tip at the end(s) of the rotor for spinning. A third gas stream directed at the rotor is used in cryogenic MAS for independent control of the sample temperature.  Spherical rotors for MAS feature only one gas stream along the equator of the sphere, which both supports the rotor with a gas-bearing and drives the spinning of the rotor. A second gas stream is also used in this setup for control of the sample temperature, and will be described

in section 2.5.

To this date, stators for spherical rotors have been developed with 3D-printing technology, which employs plastic/plastic-like material for production. This material is unsuitable for use across a wide range of temperatures, due to the thermal expansion coefficient of the 3D-printed material that results in deformation at cryogenic temperatures and loss of stable spinning. As cryogenic temperatures are necessary for DNP, and stable spinning necessary for reliable, well-resolved solid-state NMR

spectra, a stator that can spin stably across a wide range of temperatures is required. The design that we describe in this manuscript is produced in a glass ceramic (Macor®), more suitable for cryogenic application, and takes advantage of fluid flow simulations to optimize spinning stability. Combined with further 3D-print-based designs for temperature stability and magic-angle adjustability, DNP experiments are performed, achieving $^{1}$H enhancements of 256 and 200 using "large volume" (124 μL and 223 μL sample volumes, respectively) 9.5 mm spherical rotors. A stable sample temperature, with 9 W of

microwave irradiation and 2.0 kHz (+/-1 Hz) spin rate, of 105 K was achievable for this design.

**2 Cyrogenic MAS DNP Apparatus Design and Implementation**

**2.1 Stator Design**

The stator design for cryogenic spinning of 9.5 mm spherical rotors is based on the previous 3D printed designs (Chen et al., 2018; Osborn Popp et al., 2020). However, this stator is designed with the ability to use traditional manufacturing techniques

to allow the use of a material such as Macor ® for its stability and cryogenic properties which will be discussed later. The computer-assisted design (CAD) of this stator is shown in Figure 1. It includes a 9.7 mm diameter hemispherical cup (Figure 1b) which houses the spherical rotor. Fluid enters the hemispherical cup (area where the sphere is spun) via a channel with an aperture placed at the complement of the magic angle. Its entrance into the hemispherical cup is governed by a tangent plane (Figure 1b) with an opening as seen in Figure 1c and the angle of the tangent plane is detailed in Figure 1d. The tangent plane

enters the hemispherical cup smoothly guiding the fluid into it. The fluid then exits the hemispherical cup of the stator through exhaust (Figure 1a) on the far side of the stator. Manufacturing of the stator from Macor ® is performed using a 5-axis CNC. The tolerances achieved within the stator using this technique are 0.01 mm.

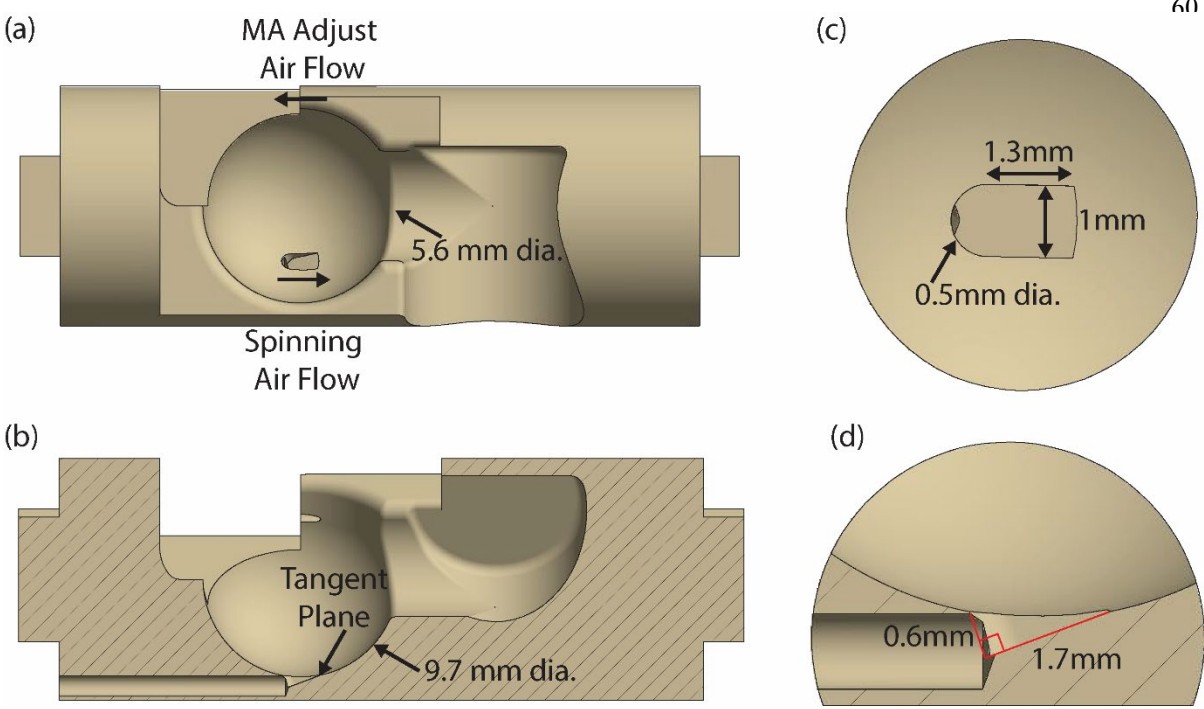

**Fig. 1. Stator Design.** (a) CAD of the stator demonstrating the flow for the spinning gas and magic angle (MA) adjust gas. The diameter of the fluid exhaust is also given. (b) CAD of the stator sliced to show the half-section of the tangent plane and the channel for spinning fluid. The diameter of the hemispherical cup is also given. (c) zoom-in from above (view (a)) highlighting the tangent plane and dimensions of the aperture. (d) zoom-in of the aperture as shown in view (b), with dimensions of the tangent plane called out.

## 2.2 Simulations for Stator and Spherical Rotor Design Optimization

Both the stator, which holds the sphere, and the sphere itself, are crucial to the fluid dynamics required for stable spinning. Two critical features that govern fluid flow in this stator are the tangent plane of the aperture and the precision of the sphericity of the spherical rotor. Previously, the important features of the stator and sphere along with their dimensions have been determined by 3D printing and rapid prototyping similar to the empirical approach used to design cylindrical rotors (Herzog et al., 2016). Recently, computational fluid dynamics (CFD) simulations have been used to explore the efficiency and design parameters of cylindrical rotors (Herzog et al., 2022, 2016). Here we are applying this approach to spherical rotors to study the critical features that govern fluid flow. CFD simulations are carried out to understand the effect that the tangent plane and sphericity of the rotor have on spinning stability. All CFD simulations are performed using Autodesk CFD 2021. The simulations are used to model fluid flow with no heat transfer and the inlet pressure is set at 1.5 bar. Meshing for the simulation is determined automatically by the program and the rotational boundary condition for the spherical rotor is set at 2.6 kHz. The

fluid was considered compressible for these simulations and they converged to a steady state, validating the conditions applied
in this model.

The first feature studied is the tangent plane which directs the main fluid stream of the stator into the hemispherical cup as can be seen in Figure 1. The absence of this tangent plane results in unstable spinning, and ejection of the sphere from the stator bowl. CFD simulations are performed to ascertain the effect of the tangent plane on spinning stability as shown in Figure 2. In the case of the tangent plane (Figure 2 left), the fluid flow has a distribution that is aligned with the aperture (forward) and
therefore the direction of spinning. When the tangent plane is removed (Figure 2 right), this distribution shifts, increasing the fluid flow and velocity normal to the aperture (normal) such that the flow aligned with the aperture (forward) diminishes. There is also an increase in flow opposite the direction of the aperture (backward). This combination results in more lift than is present with the tangent plane resulting in unstable spinning.

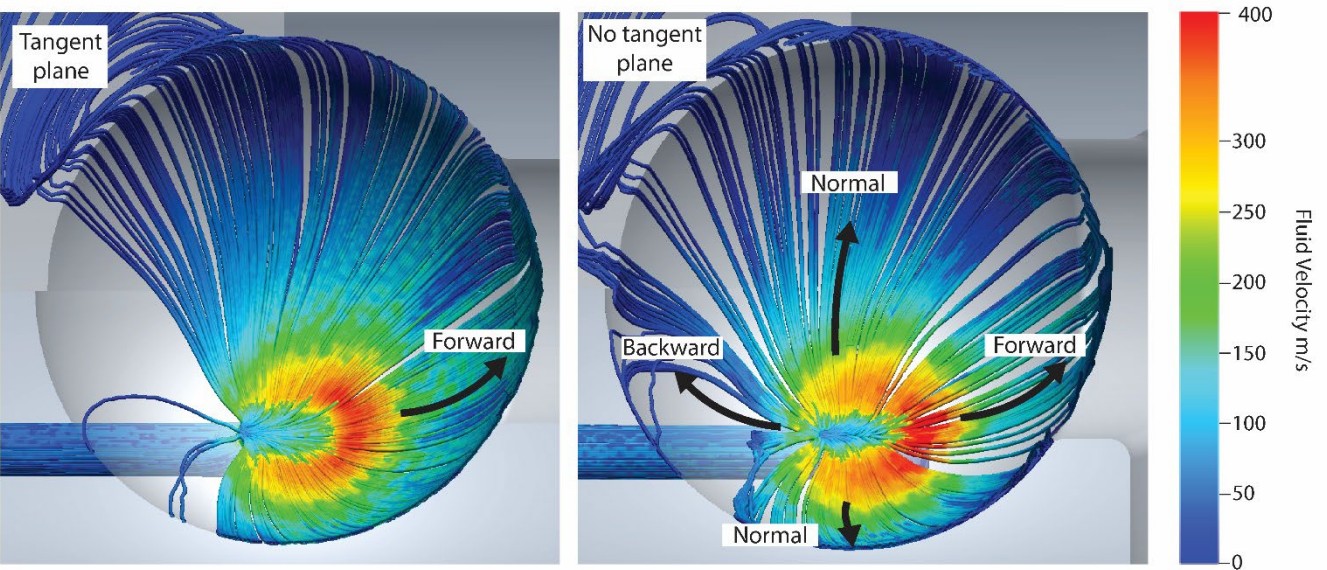

**Fig. 2. Cryogenic Stator Design.** CFD of a CAD of the stator both with and without the tangent plane. The critical features include the forward, backward, and normal fluid flows in the simulations. Changes in fluid velocity and distribution that are altered by the presence/absence of the tangent plane have been highlighted using arrows.

The second feature studied is sphericity of the spherical rotor. In previous demonstrations of spheres, rotors have been
manufactured with a cylindrical sample chamber transecting the sphere. This sample chamber is then sealed using two caps of Vespel® (Osborn Popp et al., 2020; Chen et al., 2018). However, when these spheres are spun in precisely machined Macor® stators, they exhibit poor spinning stability. Figure 3 shows a simulation of a spinning sphere with flat caps sealing the sphere chamber. This leaves a gap between the hemispherical cup and the rotor, causing turbulence and therefore spinning instability. The use of a "blind hole" sphere eliminates this issue, giving rise to stable spinning. 9.5 mm diameter grade 25 (+/- 0.0025
115   mm) sapphire spheres (Sandoz Fils SA) are used as a starting point to machine these "blind hole" spherical rotors. The 124 μL

volume rotor features a cylindrical sample chamber 5 mm in diameter and 7.2 mm in depth made by Sandoz, which does not transect the sphere making the "blind hole" (Figure 4b). This sphere is also modified in-house on a 5-axis CNC to produce the large volume (223 µL) sapphire spherical rotor sample chamber by hollowing the sphere to a thickness of 1 mm (creating a spherical-shell rotor) with a tolerance of 0.01 mm (Figure 4c). The caps, which seal the sample chamber, for both sphere designs are machined from Vespel®. The orientation of the sphere while spinning leads to microwave irradiation predominately entering the sample through the sapphire wall of the sphere, which is relatively microwave transparent at 198 GHz (Helson et al., 2018; Lamb, 1996; Afsar and Chi, 1994; Drouet d'Aubigny et al., 2010; Sahin et al., 2019).

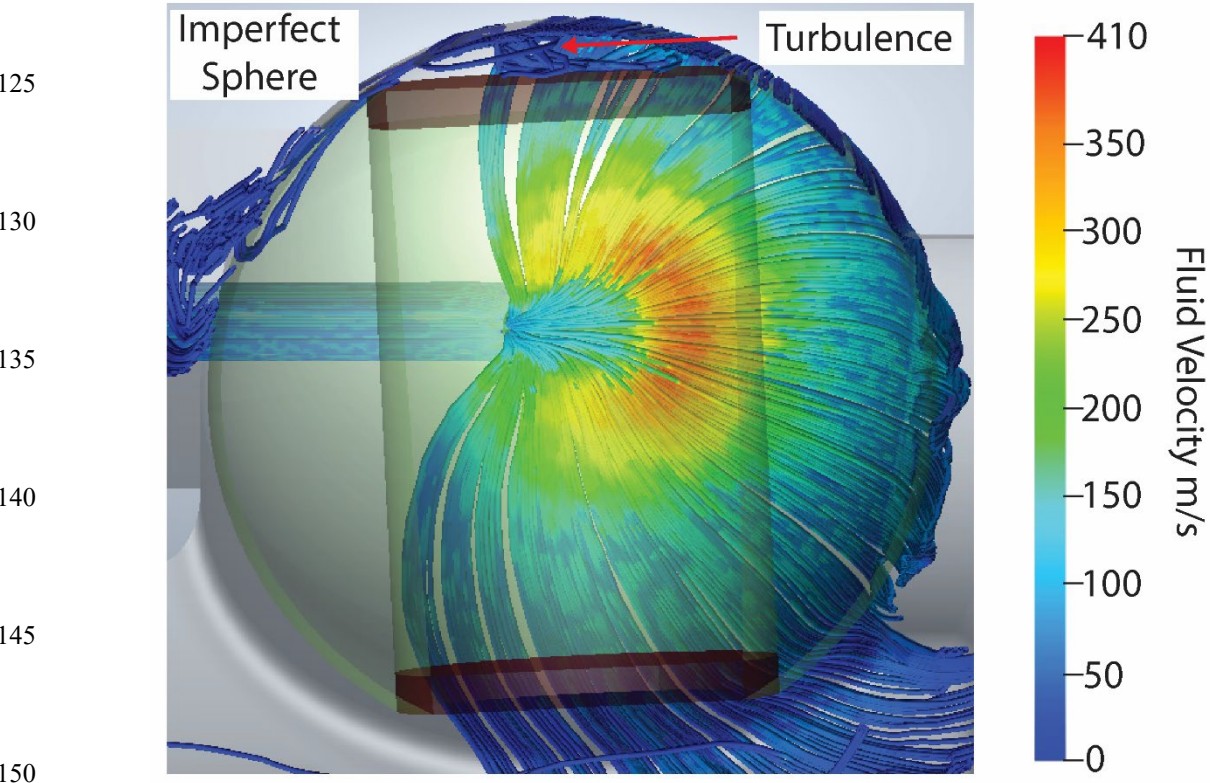

**Fig. 3. CFD of Sphere with Flat Caps.** CFD demonstrating the results of imprecise caps in the stator's hemispherical cup. The red arrow highlights the area of turbulence.

Using the results from CFD simulations, it can be seen that both the tangent plane of the stator and the precision of the sphere are critical for stable spinning in this system. Since the spinning sphere is unable to deform the Macor®, as it can the 3D printed plastic, the precision in the tangent plane, hemispherical cup, and spherical rotor are all the more necessary.

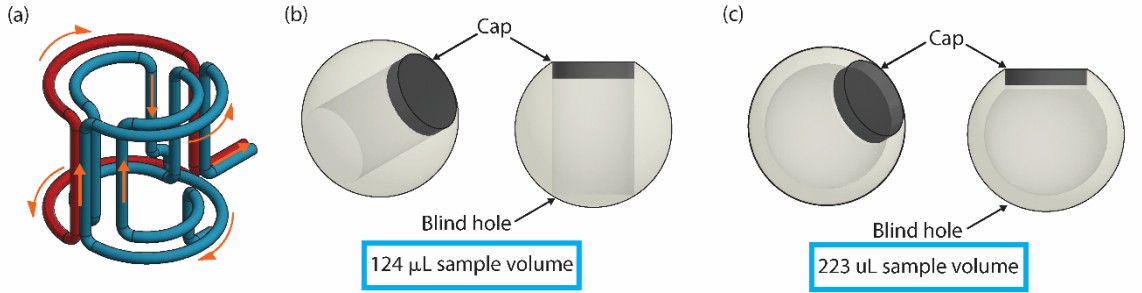

**Fig. 4. Coil and Sphere Design.** (a) CAD of a "one and a half" turn saddle coil. The blue depicts one wire and the red a separate wire. The orange arrows indicate the flow of current through the coil. (b) CAD of the "blind hole" cylindrical-chamber spherical rotor and a Vespel® cap which has a sample volume of 124 μL. (c) CAD of the "blind hole" spherical-shell rotor and a Vespel® cap which has a sample volume of 223 μL.

## 2.3 Stator Material

In addition to optimization of spinning fluid dynamics, improvements upon plastic stators are needed for stable spinning performance at the cryogenic temperatures required for MAS DNP. Previously, stators for spherical rotors were 3D printed in ABS- (acrylonitrile butadiene styrene) like plastic, which is useful for fast prototyping and proof-of-principle. However, this is not well-suited for cryogenic MAS DNP experiments because of the large thermal expansion coefficient of ABS-like plastic, as well as its softness. Attempts to use a 3D printed ABS-like plastic stator for MAS DNP result in fracturing of the printed piece at cryogenic temperatures, as well as breakdown in functionality due to mechanical wear over the course of longer experiments and repeated spin up/spin down procedures. Thus, a more robust stator was constructed using a 5-axis CNC machine (Moxley-Paquette et al., 2020), and a different material, Macor® (Corning, Inc.), which has the advantage of orders-of-magnitude greater hardness ($2.353 \times 10^9$ Pa on the Vickers hardness scale) than ABS-like plastic ($5.49 \times 10^7$ Pa). Further, the coefficient of linear thermal expansion of Macor® is $81 \times 10^{-7}$/ °C while that of the ABS like plastic is $10.1 \times 10^{-5}$/ °C, meaning Macor® will not crack or shrink significantly when cooled to the temperatures required for MAS DNP. Additionally, the combination of a Macor® stator and sapphire sphere is advantageous as the linear thermal expansion of sapphire is $88 \times 10^{-7}$/ °C, which is almost identical to Macor®. With this, both the stator and sphere shrink at the same rate when cooled. This preserves the fluid dynamics simulated and tested at room temperature when operating at the cryogenic temperatures required for DNP.

Another advantage of Macor® is its lack of protons. When using a 3D printed plastic part, there are many protons present which will show up as background in the NMR spectra. Because Macor® is a glass ceramic comprised of fluorophlogopite mica and borosilicate glass, it has no protons in its composition, greatly reducing background signal when performing proton NMR.

## 2.4 Coil Geometry

The NMR coil described here is designed to meet several requirements for MAS DNP that include radio frequency (RF) performance, sample insert/eject, and microwave access. Saddle coils have been successfully implemented in previous MAS sphere probes to meet all of these requirements (Chen et al., 2021; Gao et al., 2019a). However, in this design, a single-turn saddle coil (single saddle coil) does not result in adequate Rabi frequency for the NMR experiments, while the double saddle coil, which should improve RF performance, gives similar performance to the single saddle coil. In order to understand this, the capacitance, inductance, and self-resonance of the coils are measured using a vector network analyzer (VNA) (Rhode and Schwarz) with a range from 9 kHz to 4.5 GHz (Table 1). Each coil is connected to a known capacitor and then the resonance is measured via a loop attached to the VNA that inductively couples to the coil being measured. This number is used to calculate the inductance of the coil. The self-resonance of the coil is measured in the same manner but without a capacitor attached to the coil. This lets one calculate the capacitance with the help of the previous inductance measurement. It is important to measure the self-resonance of each coil as it is an important factor in coil performance. It arises from the fact that a coil can be thought of as being composed of an inductor and capacitor in parallel due to phenomena such as turn-to-turn capacitance. This results in each coil having a resonant frequency which is equal to $\left(2 * \pi * \sqrt{LC}\right)^{-1}$. At this resonance frequency, the impedance of the coil becomes high resulting in difficulty tuning and matching the probe along with poor RF performance. Further, above this resonance frequency, the stray capacitance between the turns of the coil is large enough to cause the coil to behave as a capacitor which also leads to poor RF performance (Massarini and Kazimierczuk, 1997; Jutty et al., 1993). Analysis of the single and double saddle coils for the 9.5 mm spherical rotor shows that the double saddle coil displays a self-resonance near the [1]H Larmor frequency (300 MHz), explaining its poor RF performance (Cook and Lowe, 1982; Roeder et al., 1984).

The solution to this self-resonance issue is a "one and a half" turn saddle coil (Figure 4a). The outer turns (one red and one blue) are electrically connected to make a Helmholtz-type section with the "double portion of the coil" and the inner turns are left as a single saddle coil (Figure 4a). This "one and a half turn" saddle coil has an inductance and impedance between that of the single and double saddle coil, keeping the self-resonance above 300 MHz. The current flow for the "one and a half" turn saddle coil is shown by orange arrows in Figure 4a. Current first flows into the inner turns that make up the inner saddle coil portion of the coil. It then splits and flows through the two outer turns simultaneously, as would occur in a Helmholtz coil, before entering the rest of the circuit. This coil provides Rabi frequencies of 63 kHz on [1]H and 60 kHz on [13]C (adequate for the NMR experiments here) using 800 W of power for each, while maintaining sample and microwave access. The 63 kHz [1]H Rabi frequency is enough to partially decouple the [1]H spins in this system and improve resolution. This is also higher than the 40 kHz obtained using a 9.5 mm cylindrical rotor and coil (Scott et al., 2018a).

| Coil Type | Inductance (nH) | Impedance @ 300 MHz (Ohm) | Self Resonance (MHz) |
|---|---|---|---|
| Single Saddle Coil | 100 | 187 | 821 |
| Double Saddle Coil | 299 | 563 | 281 |
| "One and a half" turn Saddle Coil | 157 | 297 | 432 |

**Table 1. Saddle Coil Properties.** The inductance, impedance, and self-resonance of a representative single, double, and "one and a half" turn saddle coil are listed in the table. Notice that the self-resonance of the double saddle coil is near the [1]H frequency of 300 MHz at 7T while that of the "one and a half" turn saddle coil is much higher.

The coil design here not only meets the RF performance requirements for $^1$H-$^{13}$C cross polarization NMR experiments, but also leaves a clear path for microwave transmission and sample access. Conventional probes designed for cylindrical rotors require that the microwaves for DNP pass through the solenoid coil wrapped around the sample which can reduce microwave power (Alessandro et al., 2012). This saddle coil removes the need for any impediment of microwave transmission, thus eliminating these losses. An additional benefit of this design is the flexibility of the waveguide to allow for sample insert and eject. Thus, samples can be exchanged while the NMR probe remains in a fixed position and at a constant, cryogenic temperature. This allows efficient exchange of samples and more stable DNP experiments (Barnes et al., 2009).

## 2.5 Probe-head Design

The probe-head for 9.5 mm spherical rotors is based on those designed in our group (Scott et al., 2018a) with modifications to accommodate a spherical rather than cylindrical rotor. In this study the probe-head, as seen in Figure 5, utilizes three separate gas streams: one for spinning, one for pneumatic magic angle adjust (not used in this work), and one for cooling. Nitrogen gas below 100 K is supplied by a custom heat exchanger (Albert et al., 2017) and flows through the legs, which are printed in PLA plastic using a Prusa MK3S 3D printer. The legs direct the cooling gas onto the underside of the stator. This allows for indirect cooling of the sample through contact with underside of the hemispherical cup of the stator. Directing the cold variable temperature gas onto the rotor itself, as is done with cylinders, is not possible in this case. The fluid flow above the sphere is important and is disrupted by a direct variable temperature gas. The legs also direct the spinning and pneumatic magic angle adjust gas into the stator via hollow pivots. Using pivots at the leg/adapter interface allow for unobstructed fluid-flow while retaining the stator's freedom of rotation for manual magic angle adjustment. This is done in the same manner as with cylindrical rotors. The ability to 3D print robust parts, even for cryogenic application, such as the legs and adapters for the probe-head, is advantageous as it allows for flexibility in design (Kelz et al., 2021, 2019). Shrinking in these parts is not problematic as they do not directly interface with the spinning sphere.

This probe-head design also features an axially-centered vertical waveguide to directly irradiate the sample and also serve as access for sample insert/eject. Samples are inserted by pressurizing the probe-head and then slowly depressurizing to lower

the sphere down the vertical waveguide and into the stator. Sample eject is performed by using a pump to initiate ejection, and then pressurizing the probe-head to fully eject the sphere. This last vertical section of the waveguide used for sample insert/eject is un-corrugated. The loss over this section for an un-corrugated waveguide is -2.2 dB which is similar to the -2.3 dB measured with this section being corrugated (Scott et al., 2018a).

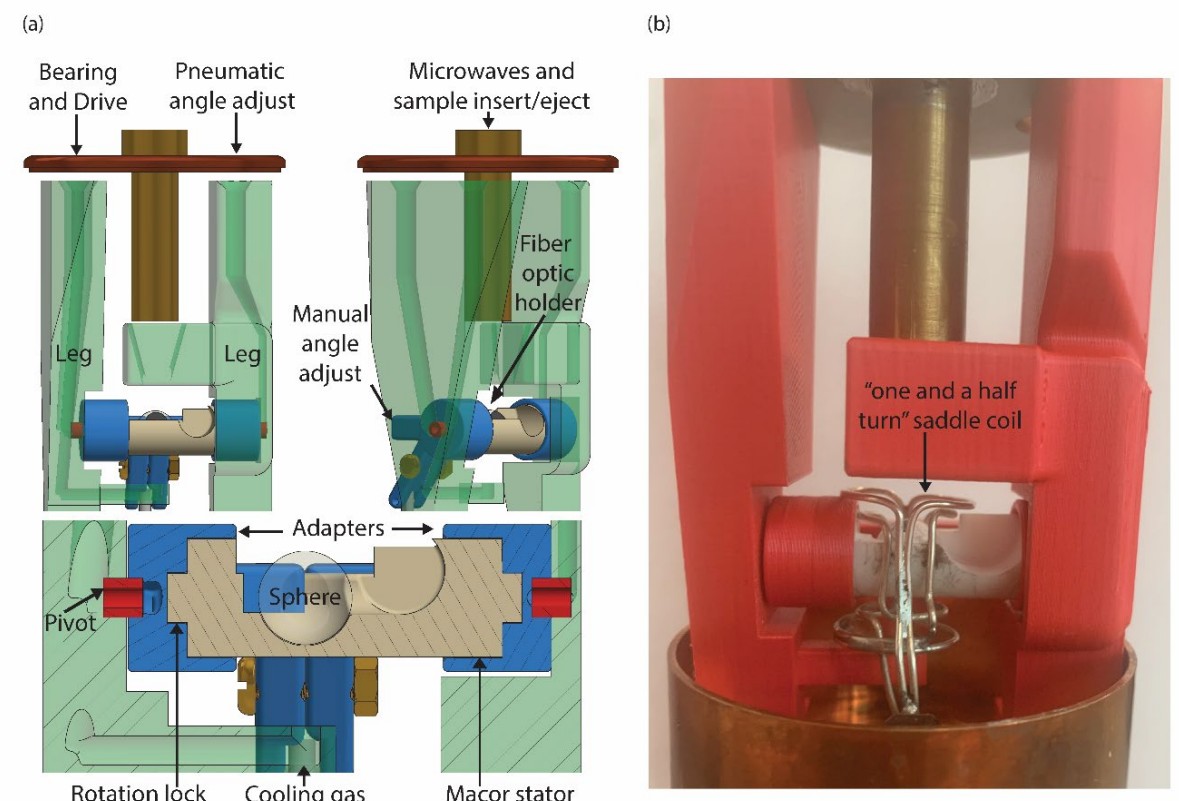

**Fig. 5. Probe-head Design.** (a) CAD of the probe-head. The gases for spinning and pneumatic magic angle adjust enter from above
265 through the 3D printed legs. Gas next travels through the pivot and into the channel in the adapter, then through the channel in the stator providing both lift and spin to the sphere. The cooling gas (variable temperature) is directed at the underside of the Macor® stator via a 3D printed channel. The center hole in the top allows microwaves to shine directly on the sample. It also doubles as an insert/eject tube for the sphere. A fiber optic holder directs and secures the fiber optics, which are used to detect the spinning frequency of the sphere. The "rotation lock" between the adapters and the stator ensures concurrent movement for
270 manual magic angle adjust. (b) Picture of the probe-head with the "one and a half" turn saddle coil included.

## 3 Cryogenic MAS and DNP Experimental Results

### 3.1 Cryogenic Spinning

Using the Macor® stator and sapphire spherical rotor described here, spinning frequencies of 3.7 kHz are achieved at room temperature using a pressure of 3 bar and a flow of 34.4 L/min, which is comparable to results obtained previously with 3D

printed stators (Osborn Popp et al., 2020). At 94 K, a flow of 28 L/min at 100 K for spinning and 30 L/min at 104 K for variable temperature is required to achieve 2 kHz spinning. Spinning stability with this design is ± 1 Hz at both room temperature and 94 K. This design also includes the ability to pneumatically adjust the angle of spinning as demonstrated in previous work with spherical rotors (Popp et al., 2021) along with the traditional mechanical magic angle adjust. In these experiments, only the mechanical adjust is utilized. Two rotors containing a cylindrical sample chamber and a single spherical-shell rotor are used in these experiments along with two copies of the Macor® stator. All of the spheres and stators used for these experiments gave similar performance.

## 3.2 DNP Spectrometer and Sample

The MAS NMR experiments are performed using a custom-built transmission line probe (Schaefer-McKay) (Scott et al., 2018a) and a Bruker console with $B_0 = 7.046$ T and carrier frequencies of 300.077 MHz for $^1$H, 75.461 MHz for $^{13}$C, and 75.192 MHz for $^{79}$Br. The samples used in this study are 124 μL and 223 μL of 4 M $^{13}$C, $^{15}$N-fully labeled urea, 20 mM AMUPol in a glassy matrix of glycerol-$d_8$/$D_2O$/$H_2O$ (60/30/10 ratio, by volume). This is the same sample previously used as a standard for MAS DNP experiments (Albert et al., 2017). A small amount (<15 mg) of KBr is encased in the bottom of the sample, separated from the urea/AMUPol. Verification of cryogenic temperatures is accomplished using $^{79}$Br spin-lattice relaxation measurements (Thurber and Tycko, 2009). $^1$H-$^{13}$C CP experiments are performed using a saturation train before longitudinal recovery delays on $^1$H spins and matching condition of 37 kHz $^1$H with 400 W and 54 kHz $^{13}$C with 350 W, and then two-phase pulse modulation $^1$H decoupling at 37 kHz with 400 W, while the sample is spinning at 2.0 kHz. The pulse powers and nutation frequencies were the same for the 124 μL and 223 μL sample volumes. DNP is performed through the cross effect mechanism with microwaves at 197.610 GHz which are generated using a custom gyrotron (Scott et al., 2018b; Gao et al., 2019b). The power of the microwaves is controlled using rotating wire grids (Thomas Keating Ltd). In these experiments the microwave power is adjusted between 1 W and 16 W. Power measurements to determine microwave power are performed using a custom water calorimeter.

### 3.3 DNP Results

Two sets of DNP experiments are performed using two different 9.5 mm spherical rotors. One rotor features a cylindrical
sample chamber (Figure 5b) with 124 μL sample volume. The second features a spherical sample chamber (Figure 5c), resulting in a 223 μL sample volume. A maximum $^1$H DNP enhancement of 256 is observed for the spherical rotor containing a cylindrical sample chamber (Figure 6a) using 8.4 W of microwave power with a sample temperature of 107 K. The power vs. enhancement curve for this sample is shown in Figure 6b, with saturation at 6.3 W. Using a 9.5 mm spherical-shell rotor a maximum DNP $^1$H enhancement of 200 is obtained using 9 W of microwave power with a sample temperature of 105 K. This
is shown in the cross effect power vs. enhancement curve for the spherical-shell rotor (Figure 7a). The DNP build-up (characterized by a time constant, $T_1$ DNP) is also recorded on the sample in the spherical-shell rotor (Figure 7b), showing the build-up of the enhanced signal with time of microwave irradiation.

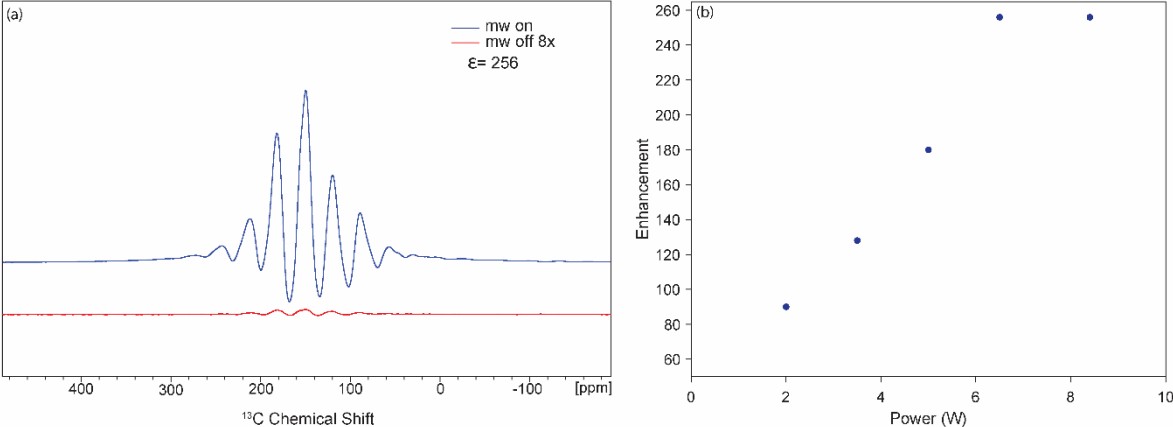

**Fig. 6. DNP Results using Small Volume Sphere (124 μL Sample Volume).** (a) DNP enhancement of 256 on $^{13}$C,$^{15}$N Urea with 20 mM
AMPUPol in 60/30/10 $d_8$-glycerol/$D_2$O/$H_2$O at a spinning frequency of 2 kHz and a temperature of 107 K. (b) DNP cross effect saturation using and enhancement vs. power curve showing saturation at 6.3 W of power.

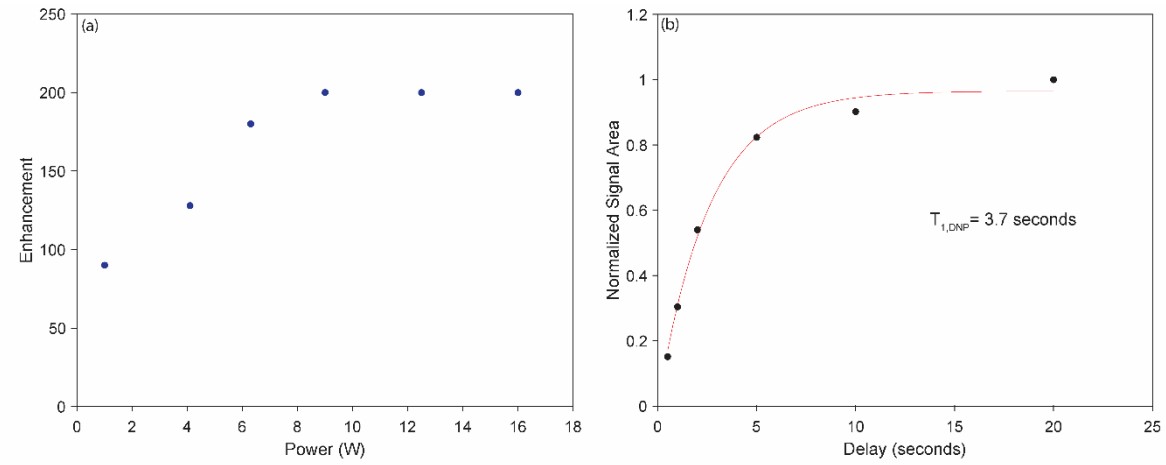

**Fig. 7. DNP Results using Spherical-shell Rotor (223 μL Sample Volume).** (a) DNP cross effect saturation using a power vs. enhancement curve showing saturation at 9 W. (b) $T_1$ DNP experiment showing the optimal $T_{1,DNP}$ of 3.7 seconds as the DNP transfer period.

The enhancement of 256 observed on the cylindrical-chamber spherical rotor matches the results from cylindrical rotor experiments with this sample (Albert et al., 2017). It is known that DNP enhancements are dependent on temperature (Rosay et al., 2010; Albert et al., 2017), spinning frequency (Mentink-Vigier et al., 2015; Purea et al., 2019), microwave power (Rosay et al., 2010), and microwave homogeneity (Rosay et al., 2010; Bajaj et al., 2007; Nanni et al., 2011). A combination of these

320 factors could explain the lower enhancement on the spherical-shell rotor. First, increasing the microwave power incident on the sample increases the sample temperature, as can be seen in the microwave power vs. temperature plot for the spherical-shell rotor in Figure 8. While this type of temperature increase is typical in conventional DNP (Purea et al., 2019), temperatures in the case of the spherical-shell rotor reach 118 K. Because temperature detrimentally affects enhancement (Rosay et al., 2010), it is reasonable to suggest that the signal comprising the final points of the curve in Figure 7b are adversely affected by

325 the increase in temperature, where lower temperatures would have allowed for higher enhancements before saturation of the cross effect. These effects would be mitigated in the case of the cylindrical-chamber spherical rotor since the thicker sapphire better dissipates the heat from the sample. It is also possible that the difference in the thickness of the two spherical rotors could lead to a difference in the efficiency of microwave transmission (Thurber et al., 2013). Another possible reason for the overall lower DNP enhancement is that the microwave homogeneity across the sample in the larger, spherical-shell rotor is

330 poorer than that for the cylindrical chamber (Bajaj et al., 2007; Rosay et al., 2010; Nanni et al., 2011). Additionally, the greater amount of sapphire that is in contact with a cylindrical surface area of sample (rather than spherical) allows for better heat transfer from the sample to the cooled sapphire, resulting in a difference in sample cooling between the two spherical rotors, which could affect the relative enhancements.

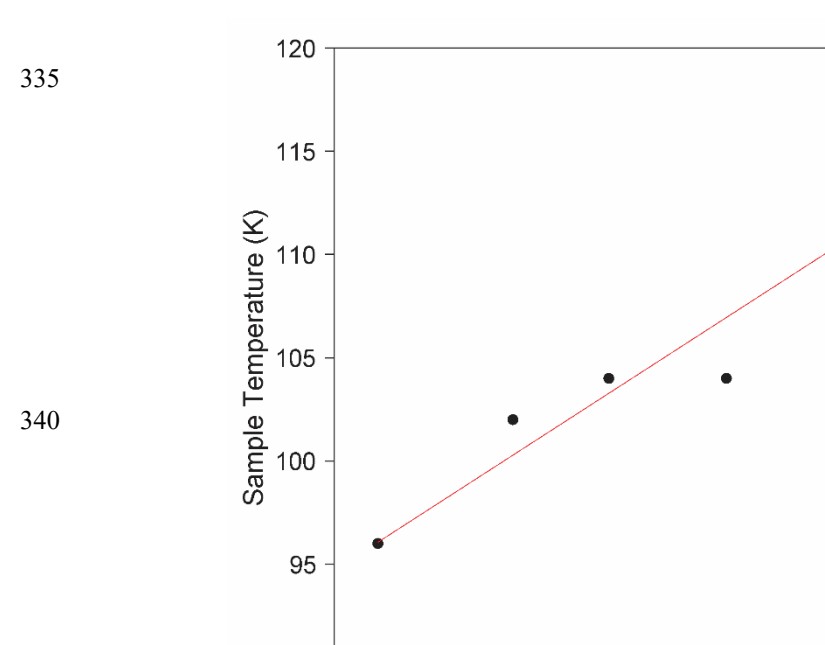

**Fig. 8. Microwave Heating Spherical-shell Rotor.** Sample temperature vs. microwave power during DNP acquisition with the spherical-shell rotor. Sample heating reaches 118 K at 16 W of microwave power.

## 4 Outlook and Conclusion

Here we describe the extension of MAS sphere technology to cryogenic MAS and its application to DNP. A Macor ® stator is produced using previous MAS sphere designs, and optimized using CFD simulations, which highlight the importance of a smooth sphere, resulting in the new design of MAS "blind hole" spheres. These two innovations in MAS sphere technology are combined with a custom 3D printed DNP probe-head. The combination of these technologies allows for the first demonstration of stable DNP experiments at cryogenic temperatures using MAS spheres.

Future improvements to this technology will enable faster spinning and colder sample temperatures. As with cylindrical rotors, smaller MAS spheres will result in higher spinning frequencies allowing for better averaging out the anisotropic interactions. Smaller rotors also provide a smaller target for microwaves, allowing for a more homogeneous effective field. The use of helium for spinning with cylindrical rotors has led to the ability to perform MAS DNP experiments below 77 K (Matsuki et al., 2015; Tycko, 2012; Thurber and Tycko, 2008). Implementing this strategy with MAS spheres will further improve the possible spinning frequencies and allow for experiments below 6 K (Judge et al., 2019; Sesti et al., 2018a, b). These cold temperatures will further increase the sensitivity of MAS DNP experiments using MAS spheres. Additionally, the adaptable

design of the probe-head and stator will aid in the implementation of MAS DNP in high-field narrow bore magnets where space is limited and allow for easier access to more unconventional experiments like EPR detection with MAS NMR.

This first demonstration of stable cryogenic operation of MAS spheres for DNP is crucial for future developments of MAS spheres. The future developments described here will allow for the application of MAS sphere technology to interesting samples for MAS NMR in the fields of biology (Gauto et al., 2021; Narasimhan et al., 2019), material science (Lesage et al., 2010; Berruyer et al., 2018; Rossini et al., 2013), and beyond.

## Author Contributions

The experiments were conceptualized by ABB and LEP. The instrumentation was designed and implemented by LEP with assistance from MU, AD, and NA. Manufacturing of the sphere and stator were performed by MU. Computational fluid dynamics simulations were carried out by LEP with assistance from TE. MAS DNP experiments were carried out by LEP with assistance from MM and NA. Initial writing of the manuscript was carried out by LEP with edits from NA. ABB supervised the experiments and manuscript preparation. All authors were included in the editing of the manuscript.

## Acknowledgements

The authors would like to thank Ronny Gunzenhauser for his advice and assistance with the 3D printing and maintenance of the equipment used in this paper. This research was supported by grant 200021_201070 from the Swiss National Science Foundation.

## Competing Interests

ETH Zürich has intellectual property protection on the inventions included in this paper. A.B.B. is on patents related to this work filed by Washington University in Saint Louis (62/703,278 filed on 25 July 2018 and 62/672,840 filed on 17 May 2018). The authors declare no other competing interests.

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
