# Peer review of "Cryogenic-Compatible Spherical Rotors and Stators for Magic Angle Spinning Dynamic Nuclear Polarization"

_Magnetic Resonance, 2023_

## Author Comment (AC2)

**Kong Ooi Tan**

**Comment 1**

*The manuscript presents the first DNP results using MAS spheres at cryogenic temperatures, which is a significant achievement towards the development of novel DNP instrumentation. I recommend the manuscript for publication subject to minor revisions (see below). Additionally, I would like to urge the authors to share the CAD files for the stators, spheres, and other relevant parts. This could encourage the community to attempt or start developing NMR instrumentation, which is an important but often neglected aspect in the field.*

**Response:** We appreciate the reviewer's comment on this matter and agree that dissemination of ideas and designs will help the field progress. As is listed in the "competing interests" section of this manuscript, there is a patent on the design of the hemispherical stator for spherical MAS. This design will soon be exploited by the company, ResonX, in product form and thus they wish to have control over the use of the exact design. The figures in the manuscript explicitly lay out the design of the stators, and now highlight specific dimensions in crucial areas of the stator. Moreover, at least one laboratory (Ilya Kaminker, reviewer 2) has been able to implement MAS spheres based on our previously published designs. As unfortunate as it is, it is the case in NMR that wider successful adaptation of technology is accomplished through commercial implementation, rather than by a handful of our instrumentation-focused laboratories implementing ideas and shaping designs. We therefore appreciate and truly understand the request to for dissemination of this specific design, but believe that this goal will be best achieved through partnering with ResonX (and other relevant companies), rather than sharing of CAD designs. That being said, we believe that the improvements to figure 1 and details shared in previous MAS sphere papers (cited within), provide an adequate source for replication of our science.

**Comment 2**

Regarding the legs (line ~47), are the FDM-printed (Prusa) parts airtight? In my experience, FDM-printed parts tend to be porous and don't maintain a good vacuum or airtight seal. I believe resin-printed (Formlabs) parts would be better suited for this application.

**Response:**

We appreciate the reviewer's suggestion and agree that the air-tightness of joints with 3D-printed pieces can be a mode of failure. These parts that were printed using FDM and the Prusa printers have been airtight; anytime there is a leak during spin testing/operation unstable spinning frequency is observed – this was not ever the case using the setup described in this manuscript. By using 100 % infill on the parts (no interior gaps) they were very robust and had no issues with air tightness was observed. However, with regards to vacuum, we have not tested these parts with vacuum applications. We did consider and try using parts from the resin-printers from Formlabs. However, when ABS-like resin was used for cryogenic application, we observe fracturing; with the fiberglass-impregnated Formlabs resin, the cryogenic performance is good, but the relatively poor resolution of this material and the post-machining required made this material less practical, given the adequacy of the PLA material printed with the Prusa. In the future, we certainly will explore printing with both Formlabs and with the Prusa printers/materials, as stator sizes and apparatus designs evolve.

**Comment 3**

Fig. 2a: It would be helpful to include the x, y, and z axes in the figure to help readers identify which 2D plane they are looking at. Additionally, it would significantly improve clarity and impact if the authors could upload the CAD files of all relevant parts to the supplementary information or an online repository.

**Response:** We thank the reviewer for identifying this potential point of confusion – we have adjusted the figure to improve the clarity and provide more information regarding crucial dimensions. In lieu of an x,y,z axis, we have provided different views and improved the figure description to provide a clearer description of the stator.

**Improvement to manuscript:** Fig. 1. Stator Design. (a) CAD of the stator demonstrating the flow for the spinning gas and magic angle (MA) adjust gas. The diameter of the fluid exhaust is also given. (b) CAD of the stator sliced to show the half-section of the tangent plane and the channel for spinning fluid. The diameter of the hemispherical cup is also given. (c) zoom-in from above (view (a)) highlighting the tangent plane and dimensions of the aperture. (d) zoom-in of the aperture as shown in view (b), with dimensions of the tangent plane called out.

**Comment 4**

Why keep saying maco is 'ceramic-like'? Macor is a glass ceramic material. To avoid confusion, I suggest replacing 'ceramic-like' with 'glass ceramic' or simply using the term 'Macor.' Furthermore, it is worth mentioning that another advantage of using Macor as the stator instead of 3D-printed plastics is that Macor is relatively 1H-free, resulting in less 1H background.

**Response:** We thank the reviewer for their proper description of the material and the additional benefit that was not called out in the original manuscript. We have made the appropriate changes, and find that this improves the paper.

**Improvements to manuscript:** The design that we describe in this manuscript is produced in a glass ceramic (Macor®)

Another advantage of Macor® is its lack of protons. When using a 3D printed plastic part, there will be many protons present which will show up as background in the NMR spectra. Because Macor® is a glass ceramic comprised of fluorophlogopite mica and borosilicate glass, it has no protons in its composition, greatly reducing background signal when performing proton NMR.

**Comment 5**

On pages 3-4, the authors mention that 'sphericity' is a critical feature for stable spinning. Can the authors comment on how it is quantified, how much sphericity is required, and what has been achieved by the manufacturer?

**Response:**

The best way to quantify what is meant by "sphericity" is the precision of the outer diameter of the sphere. This initially became an issue when there was a second cap on the sphere which was in the critical fluid flow area. Having the curvature of the cap match that of the sphere was almost impossible causing spinning instability which is also indicated in the fluid flow simulations. According to the company Sandoz the sphere is grade 25, which means that sphere's diameter tolerance is +/- 0.0025 mm. However, we did not perform tests on "higher grade" (larger diameter variability) spheres to identify what is unacceptable for stable spinning. For the internal diameter the tolerance on the machining of the hole is +/- 0.1 mm. This is only for the cylindrical hole as the sphere is hollowed out in-house.

**Improvement to manuscript:**

9.5 mm diameter grade 25 (+/- 0.0025 mm) sapphire spheres (Sandoz Fils SA) are used as a starting point to machine these "blind hole" spherical rotors.

**Comment 6**

On page 5, could the authors provide more information on how the hollowing of the sphere was performed? Was it done in-house, outsourced to a third-party machining company, or purchased as is from Sandoz Fils SA? Does the internal hollowed sphere also require high sphericity?

**Response:** We agree with the reviewer that further information on the manufacturing of the spherical-shell rotor is helpful. It has been added to the manuscript.

**Improvement to manuscript:**

The 124 µL volume rotor features a cylindrical sample chamber 5 mm in diameter and 7.2 mm in depth made by Sandoz, which does not transect the sphere making the "blind hole" (Figure 4b). This sphere is also modified in-house on a 5-axis CNC to produce the large volume (223 µL) sapphire spherical rotor sample chamber by hollowing the sphere to a thickness of 1 mm (creating a spherical-shell rotor) with a tolerance of 0.01 mm (Figure 4c).

**Comment 7**

It is known that conventional cylindrical rotors require precisely machined internal and external diameters (with a precision of a few microns). I am curious about the tolerance for the sphere.

**Response:** We thank the reviewer for their question. As was stated in the response to the reviewer's previous comment regarding the sphere, we have been using spheres with a diameter tolerance of +/- 0.0025 mm. This was "precise enough" for stable spinning. We have not explored the use of a "higher grade" to verify exactly how precise the sphere needs to be. That would be interesting to explore, as a rougher surface could potentially interact in such a way as to even catch the fluid flow for better propulsion. However, as is evident from the experimental and simulated results, deformation to the sphere can result in pockets of disrupted fluid flow that, when trapped between the sphere and the hemispherical bowl, would lead to instability.

**Comment 8**

On page 5, line 112, what is the loss tangent of Vespel? Is it lower than sapphire? Can the cap be machined from the same ceramic material (sapphire) instead of Vespel? Wouldn't using sapphire be better in terms of microwave transmission (lower loss tangent), hardness, and thermal conductivity?

**Response:**

We thank the reviewer for bringing up this point of discussion, and feel that this allows us to improve the paper by focusing more on the microwave transparency of sapphire and the geometry of the spinning sphere, relative to the incident microwaves. The loss tangent of sapphire is 0.000485, at 180 GHz and 77 K (this is discussed below in more detail in "Further explanation"). Using a cap machined from sapphire rather than a Vespel cap would be better in terms of microwave transmission. This is something that we have begun to explore. However, the benefit of the Vespel at the moment is that it is easier to insert into the sphere, and, like sapphire, it has relatively low thermal expansion, but is more malleable and does not chip, itself, or the edge of the spherical rotor upon insertion. We have actually ordered such caps, but decided on the epoxy-less Vespel caps rather than using cryogenic epoxy to secure the cap. It would be of interest to continuing exploring this option, as sapphire caps would improve microwave transmission (depending on the sphere geometry/cap diameter while spinning) and cooling.

This comment can be further addressed in a response to a comment by the second reviewer, where the top-down geometry of the configuration is discussed, and the fact that the microwaves actually pass through very little Vespel (mostly sapphire) is highlighted, and the manuscript adjusted to reflect this.

**Improvement to manuscript:**

The orientation of the sphere while spinning leads to microwave irradiation predominately entering the sample through the sapphire wall of the sphere, which is relatively microwave transparent at 198 GHz (Helson et al., 2018; Lamb, 1996; Afsar and Chi, 1994; Drouet d'Aubigny et al., 2010).

**Further explanation:**

The loss tangent of sapphire is 0.000485, at 180 GHz and 77 K, but depends on several factors, such as the sapphire crystal purity and orientation of the cut. These values are put into context in the newly cited manuscripts, "*Miscellaneous Data on Materials for Millimetre and Submillimetre Optics*" by Lamb, and "*Window materials for high power gyrotron*" by Afsar and Chi. The first of these citations is a very nice compilation of loss tangents and other relevant values for many different materials at different frequencies and temperatures. We are not able to provide an exact value of the loss tangent of Vespel at 200 GHz, but we cite the following papers, "*Dielectric properties of conductively loaded polyimides in the far infrared*" by Helson, et al. and "*Terahertz traveling wave tube amplifiers as high-power local oscillators for large heterodyne receiver arrays*" by d'Abigny, et al. The first of these may not be completely relevant, as the polyimide material (Vespel) is loaded with different conductive materials, but shows some transmittance at 200 GHz (although less than at lower frequencies). In the manuscript by d'Abigny, et al. they utilize Vespel lenses for focusing irradiation around 345 GHz.

**Comment 9**

In Fig. 3, there appear to be some local spots where the speed exceeds 350 m/s, which is above the speed of sound. Would this not result in a shock wave and potentially cause some issues? Can the authors comment on this?

**Response:**

Yes, there are spots where the speed exceeds 350 m/s in this simulation. This is also true in simulations done by Bruker on cylindrical rotors (Herzog et al., 2022, 2016). It is possible that this could cause a shock wave. However, in this case it doesn't seem to be a problem as the design is stable as are the cylindrical rotor designs which are simulated. These papers have been added to the references in the manuscript for the benefit of others interested in this topic.

**Improvement to manuscript:**

Recently, computational fluid dynamics (CFD) simulations have been used to explore the efficiency and design parameters of cylindrical rotors (Herzog et al., 2022, 2016).

**Comment 10**

In Table 1, could the authors comment on how the inductance was measured? Was it done using an LRC meter or an impedance analyzer? From my experience, most standard LRC meters perform measurements at 100 kHz, which may not yield accurate inductance values for NMR applications in the high MHz range. Additionally, the references provided on self-resonance are not directly relevant to NMR applications. I suggest adding the following two references: (1) 'A large-inductance, high-frequency, high-Q, series-tuned coil for NMR' by Bruce Cook and Lowe, and (2) 'NMR coils with segments in parallel to achieve higher frequencies or larger sample volumes' by Roeder, Fukushima, and Gibson. These references discuss the issues of self-resonance when making coils for large-volume samples, which is exactly what you experience here with the 9.5 mm sample. Moreover, the authors should also elaborate more (in a few extra sentences) on the origin of self resonance, and how to characterize them. Although this is known to people who have built NMR probes, not everyone in the field is familiar with this topic.

**Response:**

We thank the reviewers for the suggestions and interest in the details regarding the coil described in this paper. The inductance was measured using a vector network analyzer from Rhode and Schwarz with a range from 9 kHz to 4.5 GHz. The coil was connected to a capacitor and then the resonance was measured inductively with a loop. This number was then used to calculate the impedance and capacitance of the coil.  Thank you for the reference suggestions. These will be added to the paper. A longer discussion regarding self-resonance will also be added to the paper.

**Improvement to manuscript:**

The NMR coil described here is designed to meet several requirements for MAS DNP that include radio frequency (RF) performance, sample insert/eject, and microwave access. Saddle coils have been successfully implemented in previous MAS sphere probes to meet all of these requirements (Chen et al., 2021; Gao et al., 2019a). However, in this design, a single-turn saddle coil (single saddle coil) does not result in adequate Rabi frequency for the NMR experiments, while the double saddle coil, which should improve RF performance, gives similar performance to the single saddle coil. In

order to understand this, the capacitance, inductance, and self-resonance of the coils are measured using a vector network analyzer (VNA) (Rhode and Schwarz) with a range from 9 kHz to 4.5 GHz (Table 1). Each coil is connected to a known capacitor and then the resonance is measured via a loop attached to the VNA that inductively couples to the coil being measured. This number is used to calculate the inductance of the coil. The self-resonance of the coil is measured in the same manner but without a capacitor attached to the coil. This lets one calculate the capacitance with the help of the previous inductance measurement. It is important to measure the self-resonance of each coil as it is an important factor in coil performance. It arises from the fact that a coil can be thought of as being composed of an inductor and capacitor in parallel due to phenomena such as turn-to-turn capacitance. This results in each coil having a resonant frequency which is equal to $1 \big/ \left(2 * \pi * \sqrt{LC}\right)$.

At this resonance frequency, the impedance of the coil becomes high resulting in difficulty tuning and matching the probe along with poor RF performance. Further, above this resonance frequency, the stray capacitance between the turns of the coil is large enough to cause the coil to behave as a capacitor which also leads to poor RF performance (Massarini and Kazimierczuk, 1997; Jutty et al., 1993). Analysis of the single and double saddle coils for the 9.5 mm spherical rotor shows that the double saddle coil displays a self-resonance near the [1]H Larmor frequency (300 MHz), explaining its poor RF performance (Cook and Lowe, 1982; Roeder et al., 1984).

**Comment 11**

On page 8, section 3.1, if available, could the authors provide a histogram of the spinning frequency for a certain time interval? Additionally, could the authors comment on the reproducibility of the results? It would be valuable to know if the authors machined only one sapphire rotor and one Macor stator, or if such apparatus can be successfully replicated in 1 out of 3 or 10 trials.

**Response:**

We thank the reviewers for this suggestion. Rather than including a histogram we have included KBr data below demonstrating the spinning stability of the setup. This data was not included in the original manuscript as the focus was on the cryogenic spinning and DNP. This stability was also the same as observed in the cited manuscript, *"Highly stable magic-angle spinning spherical rotors"* by Osborn Popp, et al. (Magnetic Resonance 2020). This fits, as the design of the stator is most similar to the design described in that manuscript.

In the course of this study two spheres with a cylindrical sample chamber were used. Only one spherical-shell rotor has been machined so far due to the long machining time on the 5-axis CNC. All of the three spherical rotors featured diameters within the tolerance specified (+/- 0.0025mm). Two separate Macor ® stators were machined and both had similar performance. The hardness of the material and the preciseness of the machine and the program are such that sub-millimeter features, such as bowl diameter, aperture size, etc. are more repeatable than that of the 3D-printed stators, especially after repeated usage. The time to set up the program and availability of time on a shared 5-axis CNC, and simply the time for the machine to make the part all make it a long process, relative to 3D printing; thus, we believe that 3D-printing, or a combination of 3D-printing with post-machining is still a preferential method for prototyping designs.

**Improvement to manuscript:**

Two rotors containing the cylindrical sample chamber and a single spherical-shell rotor are used in these experiments along with two copies of the Macor® stator. All of the spheres and stators used for these experiments gave similar performance.

**Further Explanation:**

The figures below include 3 separate 79Br data taken at room temperature (spinning at 1980 Hz) using the setup described in this paper. Each individual spectra is comprised of 8 summed transients and the time between them is 7.78 hours. The first figure is the three spectra vertically-offset to show the spinning stability over this long period of time (same placement of the spinning sidebands). The second figure is the same three spectra overlaid to highlight the angle stability (same intensity of the sidebands). The stability was the same across sample temperatures, down to 90 K.

[Figure]

[Figure]

**Comment 12**

Abstract: It could be useful to indicate the size of the sphere somewhere in the abstract, such as '9.5 mm' if I am not mistaken.

**Response:**

We thank the reviewers for pointing out this oversight in the abstract. We agree that having this detail in the abstract is useful.

**Improvement to manuscript:**

Here we describe the optimization and implementation of a stator for cryogenic MAS with 9.5 mm diameter spherical rotors, allowing for DNP experiments on large sample volumes.

**Comment 13**

Line 25: long relaxation 'times'

**Response:**

We thank the reviewers for bringing our attention to this oversight.

**Improvement to manuscript:**

Commonly employed DNP mechanisms in solid-state NMR rely on the relatively long relaxation times of unpaired electron spins at "cryogenic temperatures" (typically below 120 K), in combination with applied microwaves (Scott et al., 2018b; Gao et al., 2019b; Nanni et al., 2013; Barnes et al., 2012), to facilitate the transfer of polarization.

**Comment 14**

Line 29: replace 'directionally-dependent' with 'anisotropic'

**Response:**

We thank the reviewers for their suggestion. We have made the change to the manuscript and agree that it improves it.

**Improvement to manuscript:**

Improved resolution in solid-state NMR is made possible by MAS, which averages anisotropic nuclear spin interactions (Cohen et al., 1957; Andrew et al., 1958; Andrew, 1981).

**Comment 15**

Lines 45-46: Rephrase for improved clarity, such as 'one for spinning, one for pneumatic magic angle adjustment (not used in this work), and one for cooling'

**Response:**

We thank the reviewers for this suggestion to improve the clarity of the manuscript.

**Improvement to manuscript:**

In this study the probe-head, as seen in Figure 5, utilizes three separate gas streams: one for spinning, one for pneumatic magic angle adjust (not used in this work), and one for cooling.

**Ilia Kaminker**

The manuscript by Price et.al desribes the next step of the magic angle spinning spheres technology. For the first time sphere spinning was achieved under cryogenic temeratures and this setup was used to demonstrate DNP experiments with state of the art signal enhancements. As such this paper describes a significant advancement of interest to the MR community.

**Comment 1**

In lines 30-34 The conventional cylindrical rotor is MAS setup described. VT gas is not mentioned though it is ubiquitos to cool down using a separate stream of a VT gas. This allows to either keep drive and beating gasses warm to ease on the spinning or to make them cold if even lower temperature is needed. This should be mentioned and discussed on how this is different / similar to the approach presented by the authors.

**Response:**

We thank the reviewers for this suggestion. A discussion regarding the similarities and differences between the use of cooling gas in the cylindrical rotor and spherical rotor setup has been added to the manuscript.

**Improvement to manuscript:**

The conventional technique for MAS utilizes a cylindrical sample chamber/rotor and two sets of gas to support and spin the sample rotor, which features a turbine tip at the end(s) of the rotor for spinning. A third gas stream directed at the rotor is used in cryogenic MAS for independent control of the sample temperature.  Spherical rotors for MAS feature only one gas stream along the equator of the sphere, which both supports the rotor with a gas-bearing and drives the spinning of the rotor. A second gas stream is also used in this setup for control of the sample temperature.

**Comment 2**

I would urge authors to expand the section 2.1 Probe Design: For example: it is not clear what is the puprpose of the cooling gas and how if at all it interacts with the rotor. Similarly, no details about the cooling gas pressure, temperature and flow are given in section 3.1.

**Response:**

We thank the reviewers for this suggestion, and believe that with this further detail the description is much improved. The section on the probe-head has been expanded, as shown below:

**Improvement to manuscript:**

The legs direct the cooling gas onto the underside of the stator. This allows for cooling of the sample through the underside of the hemispherical cup of the stator. Directly blowing the cold VT gas onto the rotor, as is done with cylinders, is not possible in this case. The fluid flow above the sphere is important and is disrupted by a direct VT gas. The legs also direct the spinning and pneumatic magic angle adjust gas into the stator via hollow pivots. Using pivots at the leg/adapter interface allow for unobstructed fluid-flow while retaining the stator's freedom of rotation for manual magic angle adjustment. This is done in the same manner as with cylindrical rotors.

At 94 K, a flow of 28 L/min at 100 K for spinning and 30 L/min at 104 K for VT is required to achieve 2 kHz spinning.

**Comment 3**

I would suggest to expand the discussion of the CFD simulation and especially the Figure 2b. Most readers of MR a non-experts in CFD simualtions. I would suggest adding an explanation what the readers should look at on Figure 2. Perhaps make arrows that point into the crucial differences. (Similar to the arrow pointing at turbulence on Figure 3.)

**Response:** We thank the reviewers for this suggestion and agree that more details in the discussion of the CFD simulations would be helpful to the MR audience. The manuscript has been edited with the goal of providing clarity and information for those new to CFD simulations.

**Improvement to manuscript:**

Both the stator, which holds the sphere, and the sphere itself, are crucial to the fluid dynamics required for stable spinning. Two critical features that govern fluid flow in this stator are the tangent plane of the aperture and the precision of the sphericity of the spherical rotor. Previously, the important features of the stator and sphere along with their dimensions have been determined by 3D printing and rapid prototyping similar to the empirical approach used to design cylindrical rotors (Herzog et al., 2016). Recently, computational fluid dynamics (CFD) simulations have been used to explore the efficiency and design parameters of cylindrical rotors (Herzog et al., 2022, 2016). Here we are applying this approach to spherical rotors to study the critical features that govern fluid flow. CFD simulations are carried out to understand the effect that these two features have on spinning stability. All CFD simulations are performed using Autodesk CFD 2021. The simulations are used to model fluid flow with no heat transfer and the inlet pressure is set at 1.5 bar. Meshing for the simulation is determined automatically by the program and the rotational boundary condition for the spherical rotor is set at 2.6 kHz. The fluid was considered compressible for these simulations and they converged to a steady state, validating the conditions applied in this model.

The first feature studied is the tangent plane which directs the main gas stream of the stator into the hemispherical bowl (area where the sphere is spun) as can be seen in Figure 1.  The absence of this tangent plane results in unstable spinning, and ejection of the sphere from the stator bowl. CFD simulations are performed to ascertain the effect of the tangent plane on spinning stability as shown in Figure 2. In the case of the tangent plane (Figure 2 left), the fluid flow has a distribution that is aligned with the aperture (forward) and therefore the direction of spinning. When the tangent plane is removed (Figure 2 right), this distribution shifts, increasing the gas flow and velocity normal to the aperture (normal) such that the flow aligned with the aperture (forward) diminishes. There is also an

[Figure]

increase in flow opposite the direction of the aperture (backward). This combination results in more lift than is present with the tangent plane resulting in unstable spinning.

**Fig. 2. Cryogenic Stator Design.** CFD of a CAD of the stator both with and without the tangent plane. The critical features include the forward, backward, and normal fluid flows in the simulations. Changes in fluid velocity and distribution that are altered by the presence/absence of the tangent plane have been highlighted using arrows.

**Comment 4**

Some crucial dimensions are missing such as channel apertures on Figure 2a. Semisphere diameter on the stator. Other dimensions that authors deem critical for the spinning performance should be added as well.

**Response:** We agree with the reviewer that including dimensions critical to spinning performance is important for this manuscript. Figure 1 has these critical dimensions included and the new section 2.1 on stator design further addresses this issue.

**Improvement to manuscript:**

The stator design for cryogenic spinning of 9.5 mm spherical rotors is based on the previous 3D printed designs (Chen et al., 2018). However, this stator is designed with the ability to use traditional manufacturing techniques to allow the use of a material such as Macor ® for its stability and cryogenic properties which will be discussed later. The computer-assisted design (CAD) of this stator is shown in Figure 1. It includes a 9.7 mm diameter hemispherical cup (Figure 1b) which houses the spherical rotor. Fluid enters the hemispherical cup via a channel with an aperture placed at the complement of the magic angle. Its entrance into the hemispherical cup is governed by a tangent plane (Figure 1b) with an opening as seen in Figure 1c and the angle of the tangent plane is detailed in Figure 1d. The tangent plane enters the hemispherical cup smoothly guiding the fluid and has been experimentally shown to be necessary for stable spinning. Fluid then exits the hemispherical cup of the stator through exhaust (Figure 1a) on the far side of the stator. Manufacturing of the

stator from Macor ® is performed using a 5-axis CNC. The tolerances achieved within the stator using this technique are 0.01 mm.

**Fig. 1. Stator Design.** (a) CAD of the stator demonstrating the flow for the spinning gas and magic angle (MA) adjust gas. The diameter of the fluid exhaust is also given. (b) CAD of the stator sliced to show the half-section of the tangent plane and the channel for spinning fluid. The diameter of the hemispherical cup is also given. (c) zoom-in from above (view (a)) highlighting the tangent plane and dimensions of the aperture. (d) zoom-in of the aperture as shown in view (b), with dimensions of the tangent plane called out.

[Figure]

**Comment 5**

There is an excellent parapgraph on the material properties in sections 2.3. I suggest also adding a discussion on the acceptable machining tolerances. ¨

**Response:** We thank the reviewers for their suggestion. A discussion of tolerances has been added to the manuscript.

**Improvement to manuscript:**

Manufacturing of the stator from Macor ® is performed using a 5-axis CNC. The tolerances achieved within the stator using this technique are 0.01 mm.

The use of a "blind hole" sphere eliminates this issue, giving rise to stable spinning. 9.5 mm diameter grade 25 sapphire spheres (Sandoz Fils SA) are used as a starting point to machine these "blind hole" spherical rotors. The 124 µL volume rotor features a cylindrical sample chamber 5 mm in diameter and 7.2 mm in depth made by Sandoz, which does not transect the sphere making the "blind hole" (Figure 4b). This sphere is also modified in-house on a 5-axis CNC to produce the large volume (223 µL) sapphire spherical rotor sample chamber by hollowing the sphere to a thickness of 1 mm (creating a spherical-shell rotor) with a tolerance of 0.01 mm (Figure 4c).

**Comment 6**

Discussion in lines 111-113 mentions that Vespel is mm-wave transparent. How much this is relevant in the presented waveguide geometry? It appears that the irradiaion from the top results in most of the mm-wave beam going through the saphire sphere wall rather than the Vespel plug.

**Response:** We agree with the reviewer that the geometry, and thus placement of the Vespel plug while spinning is such that the majority of the microwave irradiation (at least the intense portion of the Gaussian-like $He_{1,1}$ mode beam) will pass through sapphire. We felt that it was worth calling attention to the large Vespel plug in an effort to be exhaustive, but have adjusted (see below) the manuscript to highlight the geometrical placement of the Vespel to the side, and not in the path of the microwave beam.

**Improvement to manuscript:**

The orientation of the sphere while spinning leads to microwave irradiation predominately entering the sample through the sapphire wall of the sphere, which is relatively microwave transparent at 198 GHz (Helson et al., 2018; Lamb, 1996; Afsar and Chi, 1994; Drouet d'Aubigny et al., 2010).

**Comment 7**

Line 158 - Nutation freuqencise are given for both 13C and 1H cahnnels and only one power level of 800 W is given. Was 800 W used in both 1H and 13C channels?¨

**Response:** We thank the reviewer for identifying this, and have addressed it by listing the powers and nutation frequencies used throughout the CP pulse sequence.

**Improvement to manuscript:**

This coil provides Rabi frequencies of 63 kHz on $^1$H and 60 kHz on $^{13}$C (adequate for the NMR experiments here) using 800 W of power for each, while maintaining sample and microwave access.

$^1$H-$^{13}$C CP experiments are performed using a saturation train before longitudinal recovery delays on $^1$H spins and matching condition of 37 kHz $^1$H with 400 W and 54 kHz $^{13}$C with 350 W, and then two-phase pulse modulation $^1$H decoupling at 37 kHz with 400 W, while the sample is spinning at 2.0 kHz.

**Comment 8**

Line 175 - "reflection of a mirror" - why does reflection of a mirror reduces power? A properly designed mirror will be practically losless.

**Response:** We agree that a proper mirror will be practically lossless, however in practice there is some, albeit, small loss, as the copper surface is not perfectly smooth nor without any oxidation. As this is not a crux of the argument for the design of the probehead, we feel that it is appropriate to remove the comment, so that it does not take away from the rest of the description.

**Improvement to manuscript:**

Conventional probes designed for cylindrical rotors require that the microwaves for DNP pass through the solenoid coil wrapped around the sample which reduces microwave power (Alessandro et al., 2012). This saddle coil removes the need for any impediment of microwave transmission, thus eliminating these losses.

**Aaron Rossini**

This article describes the design of a probe and stator for magic angle spinning of spherical 9 .5 mm rotors at cryogenic temperatures. Previous work from the Barnes group has described stators for spinning of spherical rotors. But, this paper describes the design of a macor stator that is compatible with MAS experiments at liquid nitrogen (or potentially cooler) temperatures. The authors also have implemented a one and half turn saddle coil that improves RF performance over previous designs. This article is likely to be of interest to the solid-state NMR and DNP community. I therefore support its publication in Magnetic Resonance.

**Comment 1**

It would be great if the authors could share as much of the material needed to reproduce the data presented in the paper. For example, coud they share CAD drawings of the stator and probe?

**Response:** We appreciate the reviewer's comment on this matter and agree that dissemination of ideas and designs will help the field progress. As is listed in the "competing interests" section of this manuscript, there is a patent on the design of the hemispherical stator for spherical MAS. This design will soon be exploited by the company, ResonX, in product form and thus they wish to have control over the use of the exact design. The figures in the manuscript explicitly lay out the design of the stators, and now highlight specific dimensions in crucial areas of the stator. Moreover, at least one laboratory (Ilya Kaminker, reviewer 2) has been able to implement MAS spheres based on our previously published designs. As unfortunate as it is, it is the case in NMR that wider successful adaptation of technology is accomplished through commercial implementation, rather than by a handful of our instrumentation-focused laboratories implementing ideas and shaping designs. We therefore appreciate and truly understand the request to for dissemination of this specific design, but believe that this goal will be best achieved through partnering with ResonX (and other relevant companies), rather than sharing of CAD designs. That being said, we believe that the improvements to figure 1 and details shared in previous MAS sphere papers (cited within), provide an adequate source for replication of our science.

**Comment 2**

The discussion of lower DNP enhancements for the spherical shell rotor. I agree that having more sample (as compared to the cylindrical cavity rotor) will likely lead to more microwave heating, poorer microwave penetration/homogeneity and lower DNP enhancements. Does the wall thickness also matter? The authors should briefly cite the work of Tycko and Thurber that talks about how the thickness of zirconia rotor walls impacts microwave transmission.
https://linkinghub.elsevier.com/retrieve/pii/S1090780712003461

**Response:** We thank the reviewer for this suggestion and certainly will add the relevant citation. As to the relative effect of microwave transmission between the thin spherical sapphire shell and the solid sapphire sphere with a cylindrical chamber, we wish to comment not on the transmission of microwaves for heating, but on the effective cooling ability of a larger amount of sapphire that is in contact with a proportionally greater amount of sample.

**Improvement to manuscript:**

It is also possible that the difference in the thickness of the two spherical rotors could lead to a difference in the efficiency of microwave transmission (Thurber et al., 2013). Another possible reason for the overall lower DNP enhancement is that the microwave homogeneity across the sample in the larger, spherical-shell rotor is poorer than that for the cylindrical chamber (Bajaj et al., 2007; Rosay et al., 2010; Nanni et al., 2011). Additionally, the greater amount of sapphire that is in contact with a cylindrical surface area of sample (rather than spherical) allows for better heat transfer from the sample to the cooled sapphire, resulting in a difference in sample cooling between the two spherical rotors.

**Comment 3**

Page 10 - " Another possible reason for the higher temperature is that the microwave homogeneity across the sample in the larger, spherical-shell rotor is poorer than that for the cylindrical chamber, resulting in an overall lower enhancement (Bajaj et al., 2007; Rosay et al., 2010; Nanni et al., 2011)."

I found this statement to be confusing as written. If the microwave homogeneity was poorer, then one would assume there is less heating. It seems like microwave homogeneity, heating and DNP enhancements are being conflated. It may be better to say something like, "The larger, spherical-shell rotor likely has worse microwave homogeneity than the cylindrical chamber rotor, resulting in an overall lower DNP enhancement."

**Response:** We thank the reviewer for this comment, and agree that it is misleading. We will take the reviewer's suggestion and append that statement to this section. Additionally, we wish to comment on the difference in cooling capability of a thin-sapphire-shelled, large volume sphere from the smaller volume sphere. Please find the improvements to the manuscript below.

**Improvement to manuscript:**

It is also possible that the difference in the thickness of the two spherical rotors could lead to a difference in the efficiency of microwave transmission (Thurber et al., 2013). Another possible reason for the overall lower DNP enhancement is that the microwave homogeneity across the sample in the larger, spherical-shell rotor is poorer than that for the cylindrical chamber (Bajaj et al., 2007; Rosay et al., 2010; Nanni et al., 2011). Additionally, the greater amount of sapphire that is in contact with a cylindrical surface area of sample (rather than spherical) allows for better heat transfer from the sample to the cooled sapphire, resulting in a difference in sample cooling between the two spherical rotors.

**Comment 4**

Can the authors comment on the RF performance and NMR sensitivity for the two different rotor designs? Do the authors see nearly 2-fold larger signal from the spherical cavity rotor as they would expect from the ratio of sample volumes? Are there are differences in RF field performance between the two rotor designs?

**Response:**

We thank the reviewer for this, and have included in section 3.2 a more explicit statement conveying the similar performance between the two rotors (below). There was an increase in signal-to-noise ratio of about 2 between the two rotors, nearly as expected due to the approximately 1.6 increase in volume. The RF homogeneity of the large volume sphere was qualitatively similar to that of the small volume sphere, but further experiments would be needed to fully characterize the relative homogeneities. This would be of interest to explore in future publications. Additionally, the nutation frequencies achieved (per Watt of power) were the same for both sample volumes, and thus the same CP conditions (and powers) were used for experiments with both rotor designs.

**Improvement to Manuscript:**

$^1$H-$^{13}$C CP experiments are performed using a saturation train before longitudinal recovery delays on $^1$H spins and matching condition of 37 kHz $^1$H with 400 W and 54 kHz $^{13}$C with 350 W, and then two-phase pulse modulation $^1$H decoupling at 37 kHz with 400 W, while the sample is spinning at 2.0 kHz. The pulse powers and nutation frequencies were the same for the 124 µL and 223 µL sample volumes.

**Comment 5**

Conclusions: " As with cylindrical rotors, smaller MAS spheres will result in higher spinning frequencies allowing for better averaging out the anisotropic interactions and greater DNP enhancements, respectively."

Do faster MAS frequencies actually lead to better DNP enhancements? I believe that faster MAS often causes the enhancement with and without microwaves to increase, but that may be because there is more depolarization when the sample is spun faster (i.e., the microwave off signal is just getting smaller). Can the authors add one or two citations for this point and/or moderate the statement?

**Response:** We agree with the reviewer, and have adjusted the statement so that it does not equate spinning frequency and DNP enhancement.

**Improvement to manuscript:** As with cylindrical rotors, smaller MAS spheres will result in higher spinning frequencies allowing for better averaging out the anisotropic interactions. Smaller rotors also provide a smaller target for microwaves, allowing for a more homogeneous effective field.